# Exploitation of Vulnerabilities: A Topic-Based Machine Learning Framework for Explaining and Predicting Exploitation

Konstantinos Charmanas [1,*], Nikolaos Mittas [2] and Lefteris Angelis [1]

1 School of Informatics, Aristotle University of Thessaloniki, 54124 Thessaloniki, Greece; lef@csd.auth.gr
2 Department of Chemistry, International Hellenic University, 65404 Kavala, Greece; nmittas@chem.ihu.gr
* Correspondence: kcharman@csd.auth.gr

**Abstract:** Security vulnerabilities constitute one of the most important weaknesses of hardware and software security that can cause severe damage to systems, applications, and users. As a result, software vendors should prioritize the most dangerous and impactful security vulnerabilities by developing appropriate countermeasures. As we acknowledge the importance of vulnerability prioritization, in the present study, we propose a framework that maps newly disclosed vulnerabilities with topic distributions, via word clustering, and further predicts whether this new entry will be associated with a potential exploit Proof Of Concept (POC). We also provide insights on the current most exploitable weaknesses and products through a Generalized Linear Model (GLM) that links the topic memberships of vulnerabilities with exploit indicators, thus distinguishing five topics that are associated with relatively frequent recent exploits. Our experiments show that the proposed framework can outperform two baseline topic modeling algorithms in terms of topic coherence by improving LDA models by up to 55%. In terms of classification performance, the conducted experiments—on a quite balanced dataset (57% negative observations, 43% positive observations)—indicate that the vulnerability descriptions can be used as exclusive features in assessing the exploitability of vulnerabilities, as the "best" model achieves accuracy close to 87%. Overall, our study contributes to enabling the prioritization of vulnerabilities by providing guidelines on the relations between the textual details of a weakness and the potential application/system exploits.

**Keywords:** text mining; exploits; fuzzy clustering; topic extraction; security vulnerabilities; machine learning

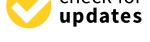



## 1. Introduction

The provision of software and system security concerns both organizations and software users as successful security breaches lead to severe incidents. Generally, incidents of this nature firstly endanger both user and system data, and secondly grant improper access to unauthorized areas. As a result, evaluating software security constitutes a practical method in assessing the existing threats, along with the potential countermeasures, and is widely studied by different perspectives [1]. One of the main perspectives is related to security vulnerabilities, described as weaknesses that are exploitable by potential cybersecurity attacks and threaten one or multiple components of a system [2]. Even though the National Vulnerability Database (NVD) (https://nvd.nist.gov/ (accessed on 11 July 2023)), which cooperates with Common Vulnerabilities and Exposures (CVE) (https://cve.mitre.org/ (accessed on 11 July 2023)), summarizes information related to disclosed vulnerabilities, the prediction and assessment of actual exploits are of great importance since many disclosed vulnerabilities are never exploited [3,4].

Consequently, prior studies focus on exploring the exploitability of security vulnerabilities by analyzing information from official databases, such as NVD and ExploitDB (https://www.exploit-db.com/ (accessed on 11 July 2023)) [3–9], and attack signatures retrieved from intrusion detection systems [4,5,8,9]. In addition, technical descriptions

that present an Exploit POC, i.e., demonstration of a concept that can lead to vulnerability exploitation, combined with clues provided by social media [9], online forums, platforms [4,5,8], and advisories [9], have contributed to prior research as well. Overall, these studies mostly leverage textual information and vulnerability characteristics to train models that classify exploitable and non-exploitable observations.

The process of studying and predicting the exploitability of security vulnerabilities is directly associated with the general field of vulnerability-patch prioritization [5]. The main goal of vulnerability prioritization is to assist organizations in maintaining system security and avoiding severe incidents by prioritizing the remediation of the riskiest security threats, i.e., providing patches to fix the vulnerabilities that are more likely to lead to actual exploits [3]. While the Common Vulnerability Scoring System (CVSS) (https://nvd.nist.gov/vuln-metrics/cvss (accessed on 11 July 2023)), which measures access and impact metrics, is a standard formula to assess the severity of security vulnerabilities [2,5], some other approaches and measures are commonly utilized as well. Therefore, the various exploit indicators are suggested as a more appropriate choice in establishing models for vulnerability prioritization as they constitute a more direct approach to identifying the vulnerabilities that are responsible for attacks against critical infrastructures [1].

The numerous text mining techniques applied in vulnerability prediction are related to both document representations that rely on word tokens and to more advanced approaches relevant to artificial neural networks [10,11]. While these techniques generally produce accurate results when combined with machine learning algorithms, e.g., Random Forest (RF) and Support Vector Machines, they do not take into consideration the potential association between the concepts included in vulnerability descriptions and the likelihood of exploitation.

To address this research gap, in the current study, we focus on extracting topics based on the textual content found in vulnerability descriptions, via word clustering, to investigate which weaknesses are strongly associated with recent exploits and most likely to be exploited in the future. The contributions of this study to vulnerability prioritization and the existing research are the topic analysis of the weaknesses and products that are related to frequent recent exploits. Further, the proposed framework, which can be easily reproduced and trained for different periods, constitutes a key point in the effectiveness of the approach as well. A characteristic that discriminates the proposed framework from the existing methodologies is the exclusive use of topic distributions, assigned to each new observation, which provide unambiguous justifications regarding the exploitability of new vulnerabilities. Moreover, while the majority of the existing approaches focus on multiple characteristics that are evaluated after a significant amount of time, our framework enables the early proactive identification of severe threats by requiring only a representative description per vulnerability.

To provide a complete framework that identifies the main concepts and assesses the likelihood of exploitation of a new vulnerability, we also make use of the topic memberships of the most recent records to train classification models. Our goal is to predict whether a newly disclosed vulnerability will possibly be exploited or not. In this regard, we collect CVE entries from the NVD data feeds to take advantage of the vulnerability descriptions and available external references. These properties help us extract topics and determine the existence of Exploit POC for each record, respectively. At this point, we note that this indicator constitutes a binary class (in our study) that characterizes each vulnerability as exploitable or not.

In the proposed framework, we initially make use of the unsupervised algorithm Global Vectors (GloVe) [12] for efficient word representations, which is a widely employed approach in similar studies with promising performance in document classification tasks [13,14]. In the next step, we apply the Uniform Manifold Approximation and Projection (UMAP) [15] dimensionality reduction technique to construct dense areas and project the extracted word representations in a two-dimensional space [16]. In the related literature, UMAP is proposed as an algorithm that enables clustering algorithms to identify coherent

clusters of word and document vectors [16–18]. In this projected vector space, we apply the standard Fuzzy K-Means algorithm (FKM), which is a soft clustering approach [19], to assign cluster memberships for the identified keywords and later for each vulnerability description. The findings from other related studies indicate that FKM is an effective approach for document classification tasks, providing higher topic coherence and accuracy than topic modeling algorithms in some cases as well [20,21]. Finally, we employ a machine learning approach [22] to train classification models by leveraging the posterior document properties produced by the proposed topic extraction approach. To evaluate the performance of the proposed framework, we also train several models based on two topic modeling algorithms, entitled Latent Dirichlet Allocation (LDA) [23] and Correlated Topic Models (CTM) [24]. The results derived from our experiments show that the proposed approach extracts interpretable topics, while at the same time, it obtains higher predictive capabilities compared to the topic modeling techniques.

The following sections of this study are organized as follows: In Section 2, we present an overview of the most related studies by discussing practical implications, outcomes, and the novel methodologies that were employed. Section 3 presents the methodology of this study, which is oriented to providing practical solutions and insights concerning our two Research Questions. The results of this study are demonstrated in Section 4 with respect to our two Research Questions and the practical implications of the proposed framework. Section 5 includes the discussion of our findings while Section 6 presents the potential limitations of this study. Finally, Section 7 concludes the paper while Section 8 presents several recommendations for future work. All major symbols used in this study are described in Appendix A.

## 2. Related Work

### 2.1. Exploitation of Security Vulnerabilities

Studying, detecting, and utilizing vulnerability characteristics is a wide area of interest involving severity indicators, types of vulnerabilities/weaknesses, as well as exploit indicators [25]. The existing approaches that explore exploit indicators mostly aim to establish machine learning models that can predict and assess the exploitability of security vulnerabilities. The related studies that will be discussed in this section are summarized in Table 1 where we should mention that the characteristic *Scoring* refers to models that assign probabilities of exploitation or predict when a vulnerability will be exploited.

**Table 1.** Characteristics of the related studies.

| Reference | Scoring | Multiple Databases | Exclusive Use of Descriptions | Topics | Period |
|---|---|---|---|---|---|
| Jacobs et al. [26] | Yes | Yes | No | No | 2016–2018 |
| Chen, Liu, Liu et al. [27] | Yes | No | No | No | Updated daily |
| Chen, Liu, Park et al. [28] | Yes | Yes | No | No | 2016–2018 |
| Sabottke et al. [9] | No | Yes | No | No | 2014–2015 |
| Almukaynizi et al. [5] | No | Yes | No | No | 2015–2016 |
| Bozorgi et al. [7] | No | Yes | No | No | 1991–2007 |
| Bullough et al. [3] | No | Yes | No | No | 2014–2015 |
| Fang et al. [8] | No | Yes | No | No | 2013–2018 |
| Tavabi et al. [4] | No | Yes | No | No | 2010–2017 |
| Bhatt et al. [6] | No | Yes | No | No | 2012–2015 |
| Charmanas et al. [29] | No | No | Yes | No | 2015–2021 |
| This study | Yes | No | Yes | Yes | 2015–2022 |

From this perspective, Jacobs et al. [26] proposed an exploitation scoring system that, instead of assigning a binary class, discriminates exploitable from non-exploitable vulnerabilities, assigns probabilities of exploitation, and provides detailed information for vulnerability prioritization. Moreover, other approaches aimed to analyze time variables to predict whether and when a security vulnerability will be exploited in the near fu-

ture [27,28]. In practice, frameworks of this nature offer practical insights into vulnerability assessment and prioritization tasks.

Furthermore, prior research supports the fact that multiple sources should be investigated for the proactive identification of cyber threats as individual websites do not cover all detected weaknesses in time. In this regard, most studies explore the main characteristics of security vulnerabilities by analyzing the NVD data feeds and turning to other sources to clarify their exploitability. Apart from the NVD, recent studies, discussed below, explored additional sources that contain useful information related to security vulnerabilities, with most of them being oriented to software security. Sabottke et al. [9] selected features from a large pool of different sources including NVD, Open Sourced Vulnerability Database (https://en.wikipedia.org/wiki/Open_Source_Vulnerability_Database (accessed on 11 July 2023)) (OSVDB), Twitter, ExploitDB, security advisories, and Symantec's protection systems (https://www.broadcom.com/products/cyber-security/endpoint (accessed on 11 July 2023)) in order to extract information about the exploitability of a vulnerability. Many studies evaluate similar information while the most important features are related to vulnerability descriptions [3–5,7–9], Exploit POCs [3–9], attacks in the wild [4,5,8,9], severity metrics [3–8], product information [3,6–9], disclosure and patch information [7], as well as weakness and attack categories [3,6–9].

The employed word and document representations, derived from the vulnerability descriptions, include both standard approaches, like Term frequency–Inverse document frequency $(Tf - Idf)$ and bag of words (bow) [3,5,7], as well as advanced approaches like document and word embeddings [4,8]. Generally, the most employed classifiers in exploit prediction are Support Vector Machines [3–7,9], Random Forest [4–6], Gradient Boosting Machines [8], Logistic Regression [5,6], and Naïve Bayes [5,6]. In general, the various vulnerability descriptions, both from official and unofficial sources (social media, forums), include details on several characteristics—e.g., severity, product, weakness type, attack and exploit details—thus indicating that the textual features offer important knowledge to vulnerability assessment and prioritization [25].

Regarding this, Tavabi et al. [4] presented a framework that leverages relevant mentions in the dark web to project vulnerabilities in embedding spaces, hence proposing an alternative approach for the prediction of exploits in contrast to many of the existing studies that rely, mostly, on official data sources. By combining the extracted document vectors with machine learning classifiers, this framework outperformed the aforementioned standard approaches (bow, $Tf - Idf$). In an attempt to boost the predictive power of the models, three additional features, describing characteristics from ExploitDB, CVSS indicators, and evidence from expert blogs, were also taken into consideration to enhance this data-driven approach.

In a similar manner, Fang et al. [8] trained models based on the FastText machine learning framework [30], which is a framework that supports both supervised and unsupervised tasks, to train word and document embeddings to predict the exploitability of security vulnerabilities. This study proves that custom data-driven approaches can outperform baseline models, which are built to satisfy general purposes and not specific goals, thus inspiring future research attempts to optimize state-of-the-art algorithms. Overall, the mainly discussed weaknesses are related to remote code execution, privilege escalation, cross-site scripting (xss), buffer overflow, denial of service, SQL injection, directory traversal, and security bypass [6,9].

### 2.2. Cluster Analysis and Word Embeddings

Algorithms for obtaining word embeddings, such as word2vec [31] and GloVe, are approaches that achieve efficient word representations allocated identically in a vector space. The effectiveness of word embeddings in projecting keyword relationships as well as their performance in topic modeling [32,33], clustering [34], and document classification [35,36] consistently inspire researchers to propose methodologies that summarize word vectors into detailed structures reflected on textual topics [37,38]. In general, the effect, evaluation,

and selection of the different techniques in cluster analysis and topic extraction vary in relevant experiments [39,40]. Some of the standard methodologies that utilize word vectors for topic extraction are the Gaussian Mixture Models (GMM) [32,41], Density-Based Spatial Clustering of Applications with Noise (DBSCAN) [13,42], hard and Fuzzy K-Means [34,43], and more complex approaches that are based on textual similarities [40].

Usually, clustering methodologies combine word and document embeddings by handling with dimensionality reduction techniques, which show high performance on mixed data. Apart from UMAP, prior studies also make use of the t-distributed stochastic neighbor embedding (t − SNE) [44] and other algorithms to effectively capture semantics in a low-dimensional space [14,45]. Angelov et al. [42] proposed a framework entitled top2vec, which is formed by a learning method that relies on word and document vectors, UMAP, and DBSCAN [46] to discover document clusters that are interpreted into topics. The corresponding study and methodology produced more representative information for multiple datasets than baseline probabilistic topic modeling methods, e.g., LDA. Similarly, Mohammed et al. [13] combined GloVe with DBSCAN to capture the semantic context in various datasets and surpassed a more standard structure based on Tf − Idf and K-means.

A crucial step towards the approaches that rely on word vectors is the choice of topic and document representations that are based on these word representations. One simple but efficient approach for document representations is to assign the mean word vector of the words that are included in each document [47,48], which is an approach that has provided a satisfactory performance in clustering tasks [37]. Some alternative document representations are based on the Tf − Idf [37,49,50] weighting scheme or additional neural network layers that aggregate word vectors [35,36].

Fuzzy clustering is related to partitioning algorithms, where each point is not assigned exclusively to a specific cluster but is assigned to multiple clusters with positive degrees of confidence [51]. D'urso and Massari [52] reviewed the variety of applications and data structures of fuzzy clustering and discovered the usage of these methodologies in both textual information and other data structures. Multiple studies that are relevant to our approach employ a fuzzy variation of the standard FKM algorithm [20] to discover clusters and topics from textual information [43,53]. While the aforementioned methodology constitutes the baseline for the development of expansions [54], some different clustering algorithms are based on mixture models (e.g., GMM) [55], distance metrics [56], and other approaches that establish efficient structures for various types of data [52,57].

## 3. Methodology

In this section, we present in detail the proposed approach along with its components serving two main tasks that are related to *Topic Extraction* (Section 3.2) and *Classification Models* (Section 3.3). In the first phase, we propose a methodology aiming at the extraction of the main themes derived from vulnerability descriptions contributing, in turn, to the identification of the more or less exploitable topics. The second phase includes all the necessary procedures that we follow in order to train classification models based on the topics extracted in the previous phase. The main parts of these two phases are summarized in Figure 1.

Briefly, we first collected the publicly available CVE data feeds from NVD and applied the necessary procedures to clean and form the datasets of this study (*Data Collection and Preprocessing*). Next, we deployed Natural Language Processing (NLP) techniques to transform the retrieved descriptions into suitable data structures for the later steps of our framework (*Text Preprocessing*). Hence, we utilized the processed descriptions and propose an approach for topic extraction based on GloVe, UMAP, and FKM (*Keyword Clustering*), while also investigating its effectiveness by training models with two baseline topic modeling approaches to compare with (*Topic Modeling*). Finally, we assigned posterior cluster memberships to the documents to clarify the more or less exploitable topics through the coefficients extracted from a Generalized Linear Model (GLM) [58].

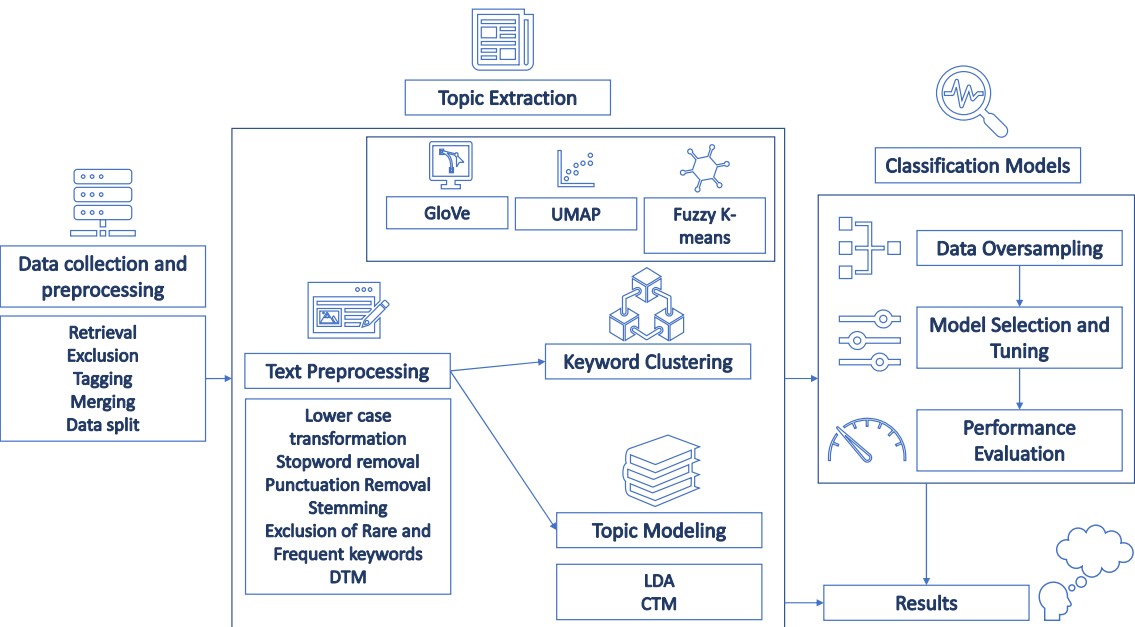

**Figure 1.** Main steps of the study's experiments.

Moreover, to fit classification models based on the cluster and topic memberships extracted in the previous task, we initially applied a data oversampling method to establish a balanced training dataset (*Data Oversampling*). Moreover, we selected two machine learning algorithms to fit classification models (*Model Selection and Tuning*). Finally, we calculated multiple performance metrics to evaluate the fitness of the trained models (*Performance Evaluation*) and the effectiveness of the proposed approach in predicting exploits.

The general motivation behind our study is to address the challenge of distinguishing exploitable types of vulnerabilities and products, and to provide information on emerging threats. In addition, we further aim to propose a complete framework that can be used for other similar tasks as well, i.e., datasets that contain textual information, which is associated with scores or classes, e.g., user reviews. Overall, the proposed methodology and the corresponding findings of this study aim to answer the following Research Questions (RQs):

RQ$_1$: Which topics of security vulnerabilities are frequently associated with recent exploits? This RQ is dedicated to providing answers on the potential relationships between the textual descriptions, expressed by topics, and the exploitability of vulnerabilities as indicated by the recent Exploit POCs (2022 records). Hence, we developed a framework that assigns a specific mixture of topics and an exploitability indicator (ranging from 0 to 1) to security vulnerabilities, as estimated by a trained GLM. The respective findings discover specific characteristics that are frequently associated with exploits while the framework can be used as a basis in vulnerability prioritization by explaining the exploitability of future vulnerabilities.

RQ$_2$: Can textual topics predict/explain the exploitability of security vulnerabilities? Although RQ$_1$ helps us assign exploitability indicators to security vulnerabilities, each threat is usually characterized as exploitable or non-exploitable with a binary class rather than a probability. In this RQ, we aim at training classification models to predict whether a vulnerability is linked with an Exploit POC or not. The motivation behind this goal is to determine if the textual descriptions and the proposed approach can establish effective classification models for future use. Also, through the answers provided for this RQ, we aim at strengthening our findings concerning the mutual characteristics of the vulnerabilities that are exploited.

### 3.1. Data Collection and Preprocessing

For the purposes of this study, we focused our analysis on NVD, as we previously discovered a strong association between the descriptions and the existence of Exploit POCs linked to each vulnerability [29]. Thus, we collected CVE records from NVD data and, more specifically, retrieved all the vulnerability records from 2015 to 2021 to fit the aforementioned topic extraction models. This collection period corresponds to the typical collection periods in related studies, which mostly retrieve data covering a range of two to six years. We also collected CVE records from 2022, until 18 July, to establish an additional dataset with more recent CVE entries that are used to reveal the current most exploitable threats, through the topics extracted from the previous datasets, and train classifiers to discover future threats. By fitting classification models with older records, e.g., from 2015, we would not succeed in addressing the current threats but the threats that have been identified as exploitable across the years, which probably do not represent the current state of vulnerability exploitation.

Moreover, we conducted a data cleaning stage to prepare the data for further analysis. Generally, NVD holds records that are excluded for various reasons, e.g., incorrect id, overwritten or duplicate records; as a result, we excluded them as well. Additionally, we excluded entries that were not yet assigned with a CVSS indicator. Next, we identified the existence or absence of Exploit POCs by inspecting the available reference tags from the retrieved dataset (2022 CVE records), since this feature is, in fact, the target class in our experimental setup. In more detail, we characterized a record as exploitable when at least one of its references (on external sources) was assigned with the tag *EXPLOIT* and discriminated the remaining records as non-exploitable. The corresponding tag symbolizes a mention or analysis of an Exploit POC included in the content of the associated references. The last step in the data preprocessing stage encompasses the merging of registrations sharing the same description. In this phase, we merged these records into a single one and linked them to Exploit POC when at least one record of the merge was characterized with the aforementioned tag. In Figure 2, we present an example of the textual description of a security vulnerability (a) and its external reference that includes an Exploit POC (b).

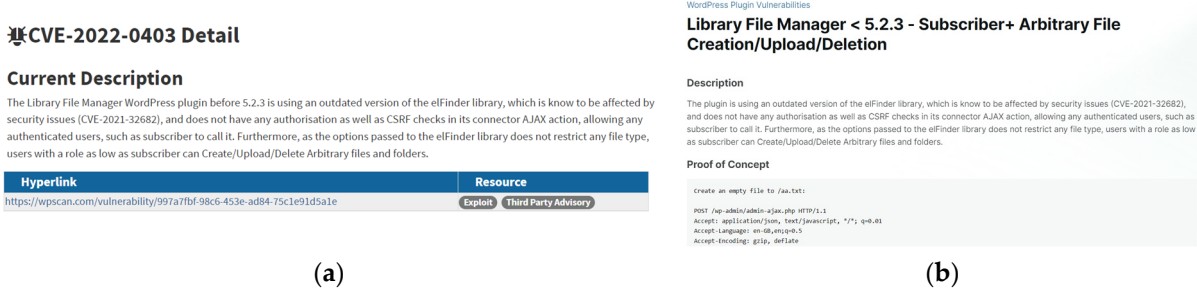

(**a**)                                         (**b**)

**Figure 2.** (**a**) Subset of the main information of a vulnerability entry. (**b**) Snippet of the content included in the external reference.

To summarize, the first dataset (2015–2021) is formed by 96,594 records after the cleaning preprocessing steps, while the dataset containing the more recent records (2022 CVE records) encompasses 8210 registrations, where 4716 are not linked with Exploit POC and the rest (3494) are associated to at least one. To train and evaluate classification models, we finally split the later-mentioned records (2022 CVE records) into training (70%) and testing (30%) datasets with respect to the target class (Exploit POC).

### 3.2. Topic Extraction

In this section, we present all the procedures that help us extract the desired knowledge from the topic and cluster models. Section 3.2.1. presents all the preprocessing techniques that were employed to transform the initial vulnerability descriptions into the appropriate structures. Furthermore, Section 3.2.2. describes the key characteristics and tuning

procedures of the two topic modeling algorithms that help us compare the performance of the proposed framework. Finally, in Section 3.2.3. we provide information on the key characteristics of the three algorithms (UMAP, GloVe, and FKM) that form the proposed framework. We should mention that the various parameters of these algorithms were tuned with respect to the employed evaluation metrics, see Section 3.2.2.

### 3.2.1. Text Preprocessing

In the current section, we describe the applied NLP procedures that assist us in achieving efficient word and document representations [59]. To eliminate keywords with identical characters, we first applied lowercase transformation to each keyword. Moreover, we removed punctuation and digit characters from the descriptions, avoiding the examination of keywords that do not carry semantics. After these two transformations, we tokenized the remaining keywords. Furthermore, we removed general terms called stopwords that surround neutral content as well. Finally, we proceeded to stemming procedures, where each token is transformed to its root context, leading us to further match the keywords with the same meaning.

Consequently, we constructed a Document-Term Matrix (DTM) by proceeding with these stemmed tokens. Also, after experimentation, we decided to exclude rare along with frequent keywords that occurred in less than 0.2% and more than 50% of the records to boost the performance models by ignoring insignificant terms. Thus, the final DTM includes 1061 keywords, which is indeed a standard type of document representation for the majority of topic modeling algorithms [60]. Also, we should mention that the same subset of keywords was utilized to establish word representations via the GloVe algorithm, which constitutes the basis of the proposed approach.

### 3.2.2. Topic Modeling

As discussed in the introductory section, we compare the performance of the proposed framework with two standard and widely applied topic modeling algorithms, i.e., LDA and CTM, which are trained using the available functionalities of the R package *topicmodels* [61]. These methodologies require specific tuning operations as they include some important prior parameters that are handled manually on each occasion. Firstly, we make use of the Variational Expectation Maximization (VEM) algorithm to tune the topic models by defining stopping criteria. Also, in this study, we select the "best" tunings for the number of topics and cluster parameters (k) using the Normalized Pointwise Mutual Information (NPMI) [62] topic coherence measure. This measure calculates the semantic coherence of a single model by evaluating the co-occurrence strength of each topic's top words. Hence, the models that carry the highest topic coherence are selected for each algorithm and are used as a basis to compare the effectiveness of the proposed approach. The key outcoming properties of these two topic models are the so-called topic distributions over documents and the word distributions over topics, which are used to train classification models and calculate the topic coherence of each model, respectively.

### 3.2.3. Keyword Clustering
GloVe

Algorithms that establish word representations aim to capture the semantic definitions and content of keywords by utilizing the textual information of a large corpus. GloVe [12] is an unsupervised algorithm for word vectors that uncovers these attributes by allocating these vectors in a way such that words that often occur in the same documents or context will have a local connectivity in the outcoming vector space as well. The main difference between GloVe with the prior technique entitled word2vec [31] is that GloVe does not rely only on local information but depends on global co-occurrence statistics derived from the investigated corpus. Thus, the inputs provided for this algorithm include the pairwise keyword co-occurrences that help us in producing the outcoming word vectors.

UMAP

In this section, we shortly present the main characteristics of the UMAP dimensionality reduction technique that leads to projecting a new low-dimensional representation of the initial data points—word vectors, in our case [15]. The practical goal of this approach is to identify the topology and relative positions of the data points in the initial vector space and project them in a lower-dimensions map. Also, this technique will allow the FKM algorithm to identify cohesive clusters by establishing dense areas. In our case, these initial data points are the word vectors extracted by the GloVe algorithm.

Firstly, the UMAP algorithm forms simplexes between the initial data points to approximate the data shape. These simplexes are created based on a manually set hyperparameter that defines the number of nearest neighbors (nn) accounted in the process. Afterwards, each point is connected with weight probabilities to each neighbor based on the distance to their closest point that creates a local radius and with zero probability to the rest of the data points. Hence, the hyperparameter nn is decisive for the upcoming output. Generally, smaller values of nn result in more local and detailed regions while higher values establish and project the global perspective of the multidimensional space.

Having the weight probabilities calculated, the algorithm projects the low-dimensional space where the data points with high-weighted edges are expected to remain in the same local region on the extracted map projection. At this point, we should mention that we make use of the cosine distance instead of Euclidean distance to identify the nearest neighbors of each data point. In our experiments, this choice leads to establishing models with higher predictive performance. Also, we should mention that this metric constitutes one of the standard alternatives to calculate the similarity/distance between word vectors.

Fuzzy K-Means

FKM is an alternative to the baseline K-means algorithm that, instead of assigning each data point to a specific cluster, assigns fuzzy partitions that represent the membership levels of each data point distributed to all established clusters. In this study, we employed the functionalities provided by the R package *fclust* [63] to fit FKM models for a predefined range of clusters. The *FKM* algorithm depends on the Euclidean distance to settle the overall posterior properties of the clusters as well as to tune the corresponding models. These tuning procedures aim to optimize the cluster partitions of the observations ($U - u_{ig}$) and the topology of the centroids ($H - h_g$) as follows:

$$\min_{U,H} J_{FKM} = \sum_{i=1}^{n} \sum_{g=1}^{k} u_{ig} d^2 \left( x_i, h_g \right) \tag{1}$$

where the predefined number of clusters is denoted by ($k$) while the cluster partitions of each observation (rows of $U$) sum to one [19]. Additionally, the data points form the $X$ ($x_i$) matrix for n observations and d denotes the Euclidean distance [63].

To optimize $k$, we first make use of the posterior properties of *FKM* to identify the top words per topic/cluster and later assess the topic coherence of the final models. The top words for each topic are assessed using the following formula:

$$KTLS_{ij} = u_{ij} \times KDP_i \tag{2}$$

where the Keyword–Topic Linkage Strength is denoted by $KTLS$ and is based on the overall Keyword Document Presence ($KDP$)—e.g., the number of documents in which each keyword occurs at least once—and the posterior cluster memberships ($U$). Thus, the top words for each topic are retrieved based on the maximum $KTLS$ values.

Finally, we select the "best" number of clusters ($k$) based on $NPMI$ and further utilize the respective posterior keyword memberships ($U$) to collectively allocate cluster memberships to the documents of the corpus. This is accomplished by assigning the

mean vector of all the keywords included in each document. Hence, we establish the Document-to-Cluster Matrix (DCM) as follows:

$$DCM = DTM \times U, DCM_{ij} = \frac{DCM_{ij}}{RS_i} \tag{3}$$

where Row Sums ($RS$) denotes the number of terms on each document derived from the row sums of the $DTM$. Another key point is that we use the properties of the settled $FKM$ model and the $DCM$ to interpret the extracted clusters into topics, with a further goal of addressing RQ$_1$ by revealing the main themes along with their exploitability. The complete flowchart of the proposed framework (Section 3.2.3) is summarized in Figure 3.

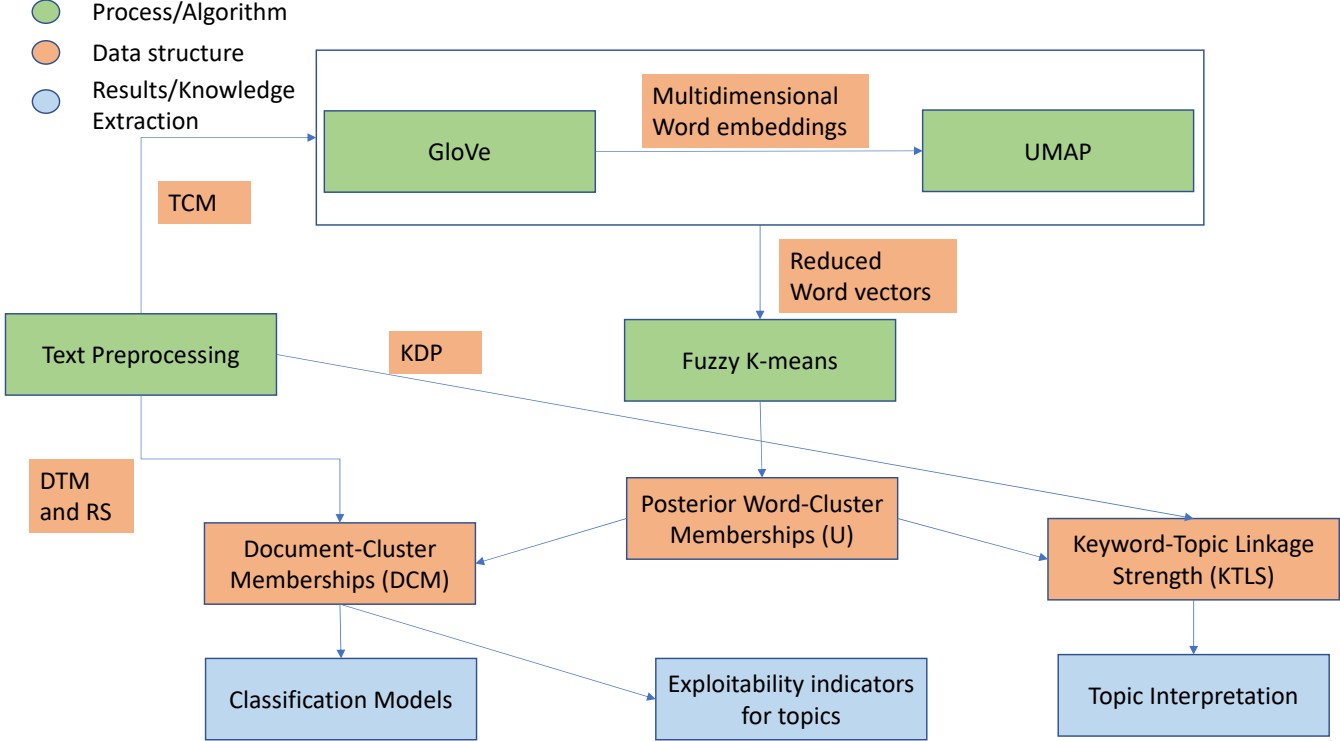

**Figure 3.** Flowchart of the proposed word clustering approach.

### *3.3. Classification Models*

The last stage of our study contains all the necessary procedures that lead to the construction and evaluation of classification models. As discussed previously, the model inputs are formed by the posterior topic distributions over documents while the output class is denoted by the binary feature that characterizes each vulnerability as exploitable or not. The main procedures of this classification stage are presented in the following sections: (i) data oversampling [64], (ii) model selection and tuning [65], (iii) performance evaluation [66].

#### 3.3.1. Data Oversampling

To balance the initial training dataset, we deployed the Adaptive Synthetic oversampling algorithm [67]. The class imbalance problem [68] occurs when an examined output feature consists of values that are mostly gathered around a few classes rather than distributed equally to every possible class. This problem leads to achieving invalid results despite a highly evaluated classification performance, in terms of accuracy measures, as many observations of the testing dataset would probably belong to majority classes as well. In these occasions, while these machine learning models aim to maximize accuracy, other performance measures like precision or recall will indicate a weak ability of the model in

predicting a minority class. Generally, oversampling algorithms produce new observations that accurately capture the characteristics of the underlying output classes and help in establishing efficient classifiers that can provide valid outcomes and support future tasks.

### 3.3.2. Model Selection and Tuning

In this section, we briefly present the steps regarding the training of classification models that involve model selection, configuration, and tuning. Our previous study [29] showed that some models achieve higher performance in predicting the exploitability of security vulnerabilities. Hence, we decided to proceed with the respective most accurate machine learning algorithms for the experiments of this study.

These algorithms are RF and C5.0, and we remind readers that the employed dataset (2022 vulnerability records) was split into training (70%) and testing datasets (30%), where the testing dataset is used after selecting the best models for each algorithm, as discussed below.

To tune the machine learning models, we employ a grid search strategy, where we construct classification models by selecting three different values for each parameter and by evaluating each combination of parameters via a 10-fold cross-validation strategy on the training dataset. RF consists of one main parameter (scale) while C5.0 consists of three (2 binary and 1 scale). Thus, for each set of inputs, we train overall thirty models (for 3 possible combinations of parameters and 10 folds) using the RF and 120 models (for 12 combinations and 10 folds) using the C5.0. Next, we evaluate each combination of parameters based on the average accuracy, as indicated by the 10-fold cross-validation strategy. Finally, by evaluating all possible parameter combinations, we select the two models that produce the highest average accuracy, one for each algorithm, to predict the exploitability of the observations belonging to the testing dataset.

### 3.3.3. Performance Evaluation

To evaluate each classifier and the effectiveness of the proposed approach, we predict the output classes of the observations, included in the testing dataset, and then analyze the findings about our framework. To do so, we rely on the confusion matrix (Table 2) that describes the number of observations in the testing dataset that belong to one of the four categories after the prediction phase. Based on this matrix, we further calculate several performance metrics that provide details on the predictive power of the employed classification models:

$$Precision : \frac{TP}{TP + FP} \tag{4}$$

$$Recall - True\ Positive\ Rate\ (TPR)\ :\ \frac{TP}{TP + FN} \tag{5}$$

$$False\ Alarm - False\ Positive\ Rate\ (FPR) : \frac{FP}{FP + TN} \tag{6}$$

$$Accuracy : \frac{TP + TN}{TP + FP + TN + FN} \tag{7}$$

$$F_1 = 2\ \frac{Precision \times Recall}{Precision + Recall} \tag{8}$$

In addition to these measures, we also make use of the Area Under the Receiver Operating Characteristics (AUC) measure. This measure is assessed from the area under the Receiver Operating Characteristics (ROC) curve, which is plotted using the TPR and the FPR measures. In binary classification, a machine learning model assesses the probability of an observation belonging to the negative class. The observations associated with probabilities that exceed a predefined threshold are classified as negative observations; otherwise, they are classified as positive observations. The ROC curve is plotted by the *TPR* and FPR values as extracted for different thresholds and the AUC denotes the area under this

plot. An example of an ROC curve and its respective AUC is presented in Figure 4. In our study, these thresholds are set to 0.5 since the training datasets are balanced through an oversampling technique. We should also mention that the values of AUC range from 0 to 1.

**Table 2.** Confusion matrix.

| | | Predicted Class | |
|---|---|---|---|
| | | Positive | Negative |
| Real Class | Positive | Observations classified correctly and belong to the positive output class (True Positives—TP) | Observations classified incorrectly and belong to the positive output class (False Negatives—FN) |
| | Negative | Observations classified incorrectly and belong to the negative output class (False Positives—FP) | Observations classified correctly and belong to the negative output class (True Negatives—TN) |

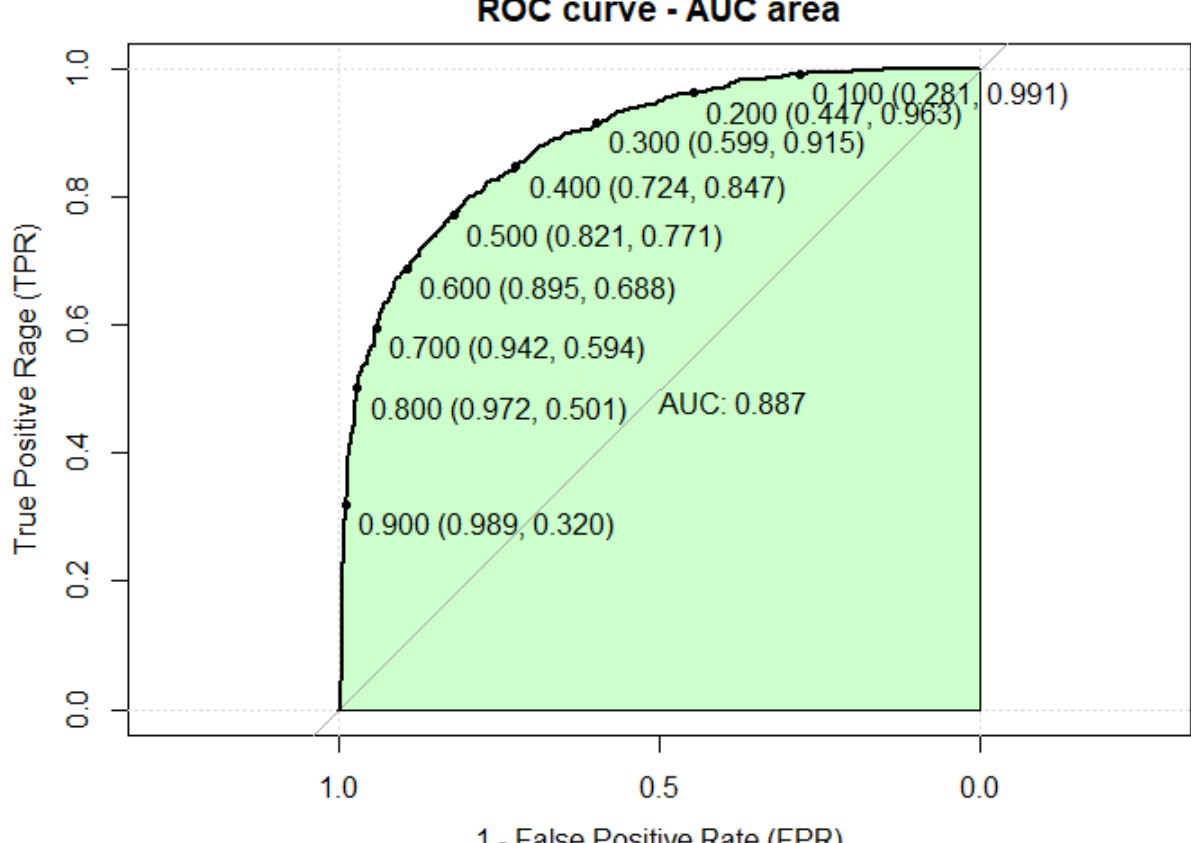

**Figure 4.** Example of an ROC curve.

In the plot above, we observe the TPR and the $1 - \text{FPR}$ values for different thresholds, where nine of them are highlighted. For example, the respective model produces an FPR equal to 0.011 and a TPR equal to 0.320 when this threshold is equal to 0.9. In general, the AUC measures the capability of the model in separating the two classes and represents the probability of a random positive observation having a smaller estimated probability of belonging to the negative class than a random negative observation [69]. In general, higher values of AUC indicate higher capabilities at distinguishing positive observations from negative ones.

This set of performance measures helps us in distinguishing whether the topic distributions over documents can effectively classify the exploitability of vulnerabilities and answer $RQ_2$. Also, we should report at this point that, in combination with NPMI, the

above measures help us evaluate the effectiveness of our approach in comparison to the previously discussed topic modeling algorithms.

*3.4. Summary*

To provide a comprehensive summary of our methodology, in this section, we describe the whole framework with inclusive and concise steps. Our goal is to guide future research in reproducing the experiments of this study or follow some distinct parts of the proposed framework that might be useful for similar tasks.

The first step is to collect all the necessary information concerning the descriptions and the exploitability of vulnerabilities through appropriate data sources, some of which are discussed in Section 2.1. In our case, we explored the NVD to find the appropriate information concerning the descriptions and the exploitability of security vulnerabilities. Also, some preprocessing procedures are necessary to transform the initial dataset into the appropriate data structures as indicated by the employed algorithms. These steps are summarized as follows:

- Retrieve information related to vulnerability descriptions and exploitability indicators;
- Apply cleaning and preprocessing procedures to the initial datasets;
- Define datasets for topic extraction (2015–2021 CVE data feeds in our case) and classification models (2022 CVE data feeds in our case);
- Employ NLP techniques to establish the DTM and TCM;

The following phase includes all the necessary procedures that should be followed to project word vectors in a low-dimensional vector space. This structure is used to assess cluster memberships to keywords and documents through the FKM algorithm. To do so, the following procedures shall be followed:

- Use TCM to train word embeddings using the GloVe algorithm;
- Employ UMAP to project these word embeddings in a low-dimensional space;
- Pipeline the outcomes of UMAP into the FKM to extract cluster memberships of keywords (U);

In turn, the DTM is used to train the topic modeling algorithms and assess cluster memberships to the documents using both DTM and U, as denoted in Equation (3). Also, some appropriate measures are used to evaluate each technique. Hence, the distinct steps are the following:

- Use DTM to train topic models;
- Calculate document memberships using the DTM and U;
- Evaluate the topic coherence of topic and cluster models using the NPMI;
- Train models for different numbers of topics as indicated by the highest NPMI of each algorithm (in our case, 24 for the proposed framework, 21 for LDA, and 10 for CTM);

By selecting the optimal model as trained using the proposed framework, the next step is to assign a comprehensive title that describes the concepts of each cluster. Also, these properties are used to explore the potential effects of the cluster membership of the documents to the exploitability of vulnerabilities. These processes are summarized as follows:

- Provide a topic title for each cluster using the top keywords and some representative descriptions;
- Evaluate coefficients that assess the potential effects of each cluster on the exploitability indicators by employing a GLM;
- Identify exploitable weaknesses and products to assist vulnerability prioritization based on the highest coefficients of the GLM;

It should be noted that the proposed framework can be adapted for other similar tasks that are related to textual information and relevant scores, e.g., user reviews. We should also clarify that, in our study, the topic extraction models were trained using the first dataset (2015–2021 CVE data feeds) while the GLM was evaluated using the second

dataset (2022 CVE data feeds). Moreover, the second dataset was used to train machine learning models. All the necessary steps that should be applied to train and evaluate machine learning models are as follows:

- Split the dataset into training (70%) and testing (30%) datasets;
- Balance the training dataset by employing an oversampling algorithm—in our case, the Adaptive Synthetic oversampling algorithm was employed;
- Select machine learning algorithms (in our case, two were selected);
- Apply a strategy that combines 10-fold cross-validation and grid search, using the training dataset, to tune the parameters of each algorithm for every set of inputs;
- Select the best parameter combinations based on the average accuracy of the respective models in the 10-fold cross-validation process;

The previous steps are applied to select the "best" machine learning models for each topic extraction (3 algorithms in our case) and machine learning (2 algorithms in our case) algorithm as well as for the three different topic parameters (3 numbers of topics in our case). These models are finally evaluated under five performance metrics that help us provide insights concerning the inclusiveness of vulnerability descriptions and the effectiveness of the proposed framework.

## 4. Results

### 4.1. RQ$_1$ Topic Assignement

In this section, we present the main results regarding the proposed keyword clustering methodology that helps us reveal the main topics of the dataset (2015–2021 data feeds). We first present the performance of the "best" three models, one for each algorithm, based on NPMI. Also, we consider exploring the performance of our approach deeper by fitting additional models for the "optimal" number of topics, one for each methodology. These numbers are indicated from the highest topic coherence evaluations, i.e., 24 for PF, 21 for LDA, and 10 for CTM. Thus, we present the NPMI evaluation of the aforementioned models to evaluate the proposed approach in terms of topic coherence. We should mention that the values produced from the NPMI measure range from $-1$ to 1. In general, values equal to 0 indicate independence between the examined words, while higher values indicate positive degrees of co-occurrence [62]. The corresponding results are presented in Table 3, where PF refers to the Proposed Framework, which combines GloVe, UMAP, and FKM.

**Table 3.** Topic coherence of constructed models.

| Model | Number of Topics | NPMI |
|-------|------------------|------|
| PF | 10 | 0.239 |
| PF | 21 | 0.246 |
| PF | 24 | 0.263 |
| LDA | 10 | 0.177 |
| LDA | 21 | 0.184 |
| LDA | 24 | 0.167 |
| CTM | 10 | 0.202 |
| CTM | 21 | 0.140 |
| CTM | 24 | 0.170 |

Moreover, we also display the UMAP projection along with the topology of the formed clusters, offering an efficient visualization that captures the relations between the detected keywords. To achieve smooth readability and better understand the relations between the main keywords of each cluster/topic, we present a graph that exclusively includes the top ten keywords of each cluster (Figure 5). We should mention that this same set of keywords was used to calculate the topic coherence of the cluster model via NPMI.

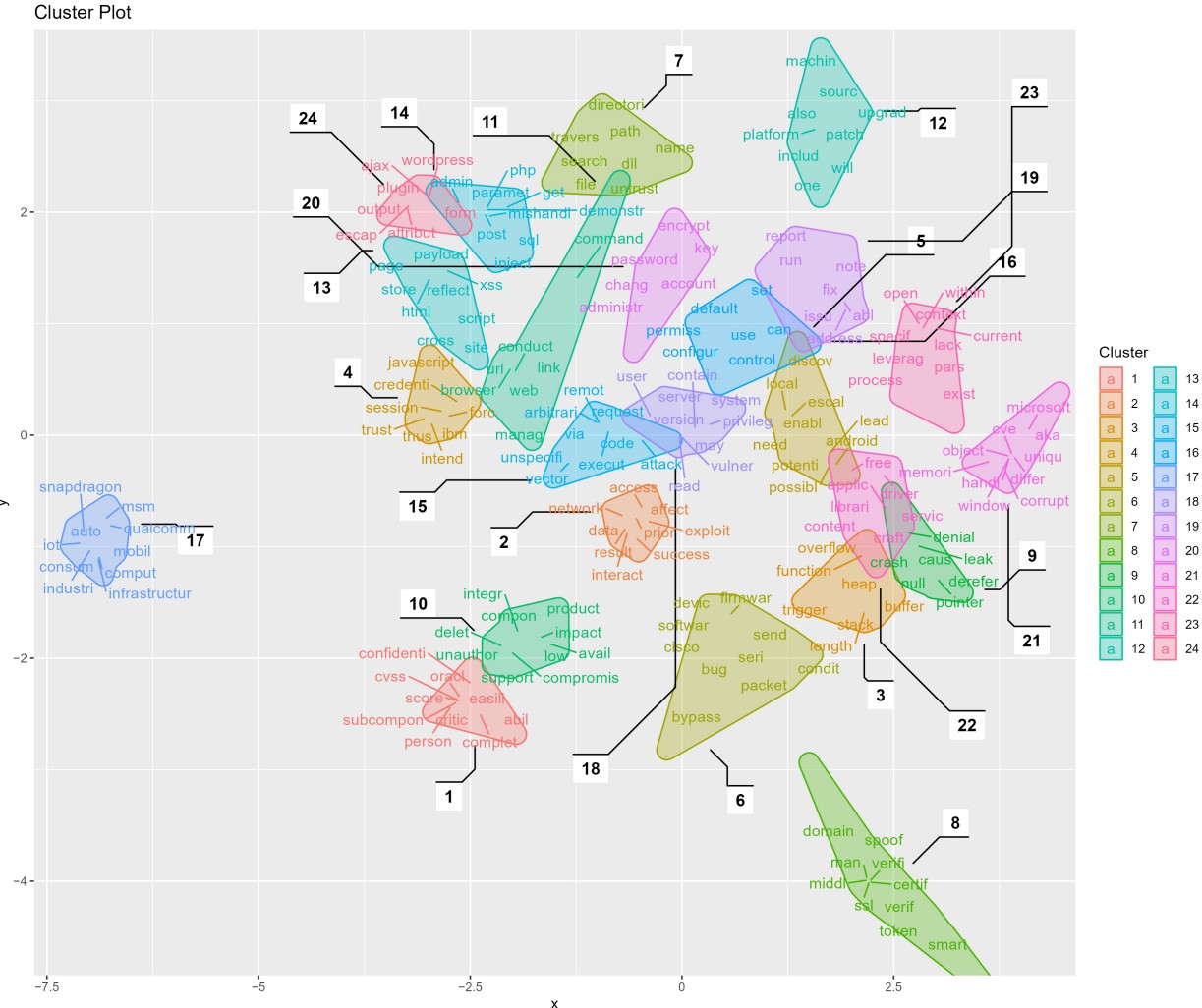

**Figure 5.** Projection and formed clusters of the proposed approach.

In addition, we present the main information of each cluster, forming a single topic, to address and answer RQ$_1$. Firstly, we define a comprehensive title for each established topic by analyzing the main keywords and some representative vulnerability descriptions (Table 4). Also, we calculate the Expected Document Frequency (EDF) of each topic in the recent CVE entries of 2022 as follows:

$$EDF_i = \sum_{j=1}^{n} \frac{DCM_{ji}}{n} \tag{9}$$

where *n* is equal to the number of documents.

**Table 4.** Main cluster/topic information.

| Cluster/Topic No. and EDF | Titles | Top 10 Stemmed Keywords |
|:---:|:---:|:---:|
| 1—0.0145 | Vulnerabilities on Oracle products with displayed severity metrics | *oracl; score; confidenti; cvss; easili; complet; subcompon; critic; abil; person* |
| 2—0.0652 | Successful attacks–exploits that affect specific data components | *affect; exploit; access; result; data; success; prior; attack; network; interact* |
| 3—0.0338 | Buffer and stack overflow | *buffer; craft; function; overflow; stack; heap; servic; trigger; denial; length* |
| 4—0.0210 | Credentials disclosure and session breaches especially on IBM products | *ibm; forc; session; credenti; intend; javascript; site; trust; cross; thus* |

**Table 4.** *Cont.*

| Cluster/Topic No. and EDF | Titles | Top 10 Stemmed Keywords |
|---|---|---|
| 5—0.0507 | Vulnerabilities on Android, especially local privilege escalation | *lead; local; potenti; possibl; discov; can; enabl; android; need; escal* |
| 6—0.0273 | Bugs on devices due to packet mishandling from protocol units, e.g., ipv, udp, tcp | *send; devic; packet; softwar; condit; bug; bypass; cisco; firmwar; seri* |
| 7—0.0344 | Path-directory traversal | *file; directori; path; travers; search; name; dll; untrust; command; attack* |
| 8—0.0100 | SSL certificates that allow spoofing from man-in-the-middle attacks | *token; certif; spoof; middl; verifi; man; ssl; smart; domain; verif* |
| 9—0.0305 | Vulnerabilities that cause system crashes and denial of service especially from null pointer dereference | *caus; denial; crash; servic; craft; pointer; function; null; leak; derefer* |
| 10—0.0294 | Unauthorized compromises on specific Enterprise product components | *compon; product; impact; compromis; support; integr; unauthor; delet; avail; low* |
| 11—0.0349 | Attacks on the web via URL redirection and malicious links | *web; manag; url; link; site; attack; browser; conduct; cross; user* |
| 12—0.0276 | Vulnerabilities with discussed affected or fixing patches especially on TensorFlow platform | *includ; will; patch; sourc; platform; also; one; upgrad; issu; machin* |
| 13—0.0501 | Cross-site scripting | *xss; script; html; cross; site; page; store; reflect; inject; payload* |
| 14—0.0652 | SQL injection and input mishandling, especially on php components | *paramet; php; inject; file; sql; get; admin; mishandl; demonstr; post* |
| 15—0.0967 | Remote code execution | *via; remot; execut; arbitrari; attack; code; request; user; vector; unspecifi* |
| 16—0.0567 | Incorrect controlling, configuring, and granted permissions | *use; can; configur; control; issu; attack; permiss; user; set; default* |
| 17—0.0051 | Vulnerabilities on Snapdragon products | *mobil; snapdragon; comput; consum; msm; infrastructur; industri; iot; auto; qualcomm* |
| 18—0.1459 | Vulnerabilities that affect specific versions and allow privilege gain | *vulner; version; privileg; attack; user; may; read; system; server; contain* |
| 19—0.0260 | Vulnerabilities with reported issues and fixes | *issu; can; note; fix; address; run; abl; discov; report; use* |
| 20—0.0302 | Hardcoded keys along with weak passwords and encryptions | *password; key; attack; administr; chang; file; user; account; encrypt; can* |
| 21—0.0404 | Vulnerabilities including discussed CVE codes, especially on Microsoft products and on vulnerabilities with memory issues | *"aka; memori; cve; window; handl; corrupt; object; differ; microsoft; uniqu* |
| 22—0.0391 | Vulnerabilities on media components and Nvidia GPU drivers, especially denial of service, use after free, and double free | *servic; applic; craft; denial; function; driver; overflow; content; librari; free* |
| 23—0.0262 | Lack of security within specific processes that attackers leverage to intervene in the related context | *context; within; open; specif; exist; lack; leverag; process; current; pars* |
| 24—0.0387 | Vulnerabilities on WordPress plugins | *plugin; wordpress; admin; page; escap; form; file; output; attribut; ajax* |

By inspecting the cluster visualization and the topic titles, we notice that topics 9 and 22 are overlapping at a high level. To distinguish the differences between these two topics, we further investigate the keywords that are closer to each cluster center. While topic 9 includes keywords of general content, we noticed that the keywords in topic 22 are mostly related to media and driver components. These keywords are the following: layer; media; mediaserv; librari; handler; mode; nvidia; intern; gpu; driver.

Meanwhile, we also reveal the current exploitability of each topic, based on the topic partitions of the 2022 vulnerability records, by evaluating the coefficients of a GLM, i.e., Binomial Logistic Regression (BLR). We remind readers that the topic partitions (inputs) range is between 0 and 1 while the Exploit POC (outputs) is a binary variable defined by values equal to 0 and 1 for POC absence and presence, respectively. In Table 5, we present the coefficients produced by GLM while Equation (10) presents the formula of this model adapted to our data and symbols. The indicator $Prob_i$ denotes the estimated probability of exploitation of the $i$-th observation/vulnerability. The GLM is fitted using the weighted least squares method, which aims to minimize the residuals of the initial data points from the estimated regression curve. This curve describes the relationships between the independent variables and the dependent variable [70].

$$Prob_i = \frac{1}{1 + e^{-b_0 + \sum_{j=1}^{g} b_j u_{ij}}} \tag{10}$$

**Table 5.** Exploitability coefficients.

| Topic No. | GLM Coefficient Estimate ($b_j$) | Odds Ratio ($e^{b_j}$) |
|---|---|---|
| 1 | −10.627 *** | 0.000024 |
| 2 | −7.283 *** | 0.000687 |
| 3 | 5.155 *** | 173.286 |
| 4 | −14.230 *** | 0.000001 |
| 5 | −7.755 *** | 0.000429 |
| 6 | −7.119 *** | 0.000809 |
| 7 | −0.988 | 0.372251 |
| 8 | −11.790 *** | 0.000008 |
| 9 | 2.200 * | 9.031704 |
| 10 | −5.950 *** | 0.002605 |
| 11 | −1.245 | 0.287895 |
| 12 | 4.0589 *** | 57.91047 |
| 13 | −2.162 ** | 0.115115 |
| 14 | 6.051 *** | 424.7054 |
| 15 | −1.285 * | 0.276587 |
| 16 | −8.134 *** | 0.000293 |
| 17 | −12.853 *** | 0.000003 |
| 18 | −6.438 *** | 0.001599 |
| 19 | −3.635 ** | 0.02638 |
| 20 | 1.367 | 3.925521 |
| 21 | −11.616 *** | 0.000009 |
| 22 | −9.0170 *** | 0.000121 |
| 23 | −9.6196 *** | 0.000066 |
| 24 | Linearly Dependent (Aliased) on Intercept | Aliased with Intercept |
| Intercept | 3.480 *** ($b_0$) | 32.444549 |

*p*-value from GLM estimations: *** $< 0.001$; ** $< 0.01$; * $< 0.05$.

The knowledge extracted from the above tables and figures helps us reveal the main topics along with their exploitability. This information constitutes the basis for addressing and providing answers to $RQ_1$. By observing the main themes and EDF, we discover that the overall prevalence of the identified topics is distributed in a balanced way on the recent vulnerabilities (2022 data feeds) as these values are not gathered around distinct themes. Also, we detect familiar types of vulnerabilities in the core of the aforementioned topics—e.g., SQL injection, cross-site scripting, and remote code execution—along with some popular products and software vendors.

According to the coefficients of Table 4, we discover that the corresponding most exploitable topics are related to buffer and stack overflow, SQL Injection, as well as WordPress plugins; at the same time, recent vulnerabilities that are associated with Snapdragon products, credential disclosure, and session breaches consist of less exploitable threats. Usually, a major issue concerning the exploitation of buffer and stack overflows is the process of memory access and overwriting that is not handled properly by many applications, making the attacked system vulnerable and easily accessible. Thus, software vendors and IT departments should focus on enabling/enhancing the security of operations that involve memory access and manipulation as these types of vulnerabilities may cause severe damage to the affected systems, e.g., sensitive data disclosure and denial of service. Regarding WordPress plugins, we believe that the reason behind the high exploitability likelihood of the associated security vulnerabilities is related to the fact that WordPress is open source. As a result, WordPress users should review the details and security of each plugin before they select the appropriate ones for their website. As for the security vulnerabilities that are linked to SQL Injection, some special characters and operations of SQL queries that are not handled properly expose the weaknesses of each database. These weaknesses may lead to the leaking of sensitive data and gaining access to unauthorized areas. SQL Injection is associated with many actual exploits due to the accessibility of online databases and the potential privileges gained from successful attacks, which, in turn, attract the interest of attackers.

Recapping, through a combination of algorithms and steps that form the PF, we established a word clustering methodology that assigns the extracted outcomes into topics and at the same time relates the analyzed documents to these topics as well. By evaluating the topic coherence and the interpretability of the extracted model, we indeed demonstrated that the PF should be regarded as a good alternative to topic modeling algorithms. Also, in contrast to the topic modeling algorithms, which analyze large collections of both documents and words, the PF discovers the relations between words and extracts the desired information significantly faster than these algorithms. Furthermore, we employ an approach that relates the extracted topics with a binary class, through a GLM, by making good use of the posterior cluster memberships of documents. In general, this approach could be useful in future studies that aim to investigate the potential effects of the textual properties on a target variable. To summarize, through the PF, we identified the main topics that encompass the textual descriptions of security vulnerabilities and further assessed indicators that relate each topic with the likelihood of exploitation of security vulnerabilities, answering effectively to $RQ_1$.

### 4.2. $RQ_2$ Exploit Prediction

Having two different supervised machine learning algorithms employed to establish classifiers for each examined model, i.e., the nine models discussed in the previous section, we overall evaluate the performance of 18 different classification models. Table 6 provides information regarding the performance measures of these models in the testing dataset. In the respective results, we should mostly emphasize the overall performance of the models that were trained based on the posterior properties produced by the PF compared with the other two topic modeling methodologies.

**Table 6.** Classification performance evaluation.

| Model | Topics | Classifier | Accuracy | Precision | Recall | $F_1$ | AUC |
|-------|--------|-----------|----------|-----------|--------|-------|-----|
| CTM | 10 | C5.0 | 0.7332521 | 0.8528864 | 0.6473498 | 0.7360386 | 0.8507613 |
| CTM | 10 | RF | 0.7945595 | 0.8369162 | 0.7978799 | 0.816932 | 0.8742083 |
| CTM | 21 | C5.0 | 0.7803492 | 0.861755 | 0.735689 | 0.7937476 | 0.870051 |
| CTM | 21 | RF | 0.8018676 | 0.8508706 | 0.7943463 | 0.8216374 | 0.8851462 |
| CTM | 24 | C5.0 | 0.7864393 | 0.8459144 | 0.7681979 | 0.8051852 | 0.8778039 |
| CTM | 24 | RF | 0.8075518 | 0.8477458 | 0.8106007 | 0.8287572 | 0.8925603 |
| LDA | 10 | C5.0 | 0.7860333 | 0.852381 | 0.7590106 | 0.8029907 | 0.8733546 |
| LDA | 10 | RF | 0.8148599 | 0.8477157 | 0.8261484 | 0.8367931 | 0.8904948 |
| LDA | 21 | C5.0 | 0.8262282 | 0.862601 | 0.829682 | 0.8458213 | 0.9061763 |
| LDA | 21 | RF | 0.8278522 | 0.8511694 | 0.8487633 | 0.8499646 | 0.9114623 |
| LDA | 24 | C5.0 | 0.8278522 | 0.8630037 | 0.8325088 | 0.847482 | 0.9136154 |
| LDA | 24 | RF | 0.8367844 | 0.8594748 | **0.8558304** | 0.8576487 | 0.9138086 |
| PF | 10 | C5.0 | 0.7941535 | 0.8229018 | 0.8176678 | 0.8202765 | 0.8635709 |
| PF | 10 | RF | 0.8124239 | 0.8420675 | 0.8289753 | 0.8354701 | 0.8866159 |
| PF | 21 | C5.0 | 0.81689 | 0.8559823 | 0.8190813 | 0.8371253 | 0.9060425 |
| PF | 21 | RF | 0.8327243 | **0.8663258** | 0.8381625 | 0.8520115 | 0.912211 |
| PF | 24 | C5.0 | 0.8282582 | 0.8547926 | 0.844523 | 0.8496267 | 0.9125877 |
| PF | 24 | RF | **0.8371904** | 0.8611111 | 0.854417 | **0.857751** | **0.9157352** |

Thus, by comparing the models with the same number of topics, we observe that the classifiers combined with the PF outperform the remaining approaches in most cases. Also, we detect that the best evaluations of the inspected metrics are mostly gathered around the PF model with 24 topics (**boldface** text). As the classification performance in all cases is better than assigning every entry of the testing dataset to a single output class (57%), we can accept that $RQ_2$ is answered positively.

Indeed, these results indicate that the descriptions provided for security vulnerabilities contain meaningful information concerning several aspects of the related weaknesses, as proven also by existing studies that were discussed in Section 2.1. By comparing the models,

which are trained for the same number of topics, we inspect that the PF outperforms the other two approaches in just one metric for 10 topics while LDA performs slightly better under the remaining evaluation metrics. As for the models that are trained for 21 topics, the PF achieves the best evaluations in four metrics, except recall. Finally, the remaining six models show that the predictive power is higher as the number of features increases. We should also mention that the PF outperforms the two algorithms in four out of five metrics for 24 topics. Overall, we detect that the LDA and the PF obtain similar evaluations under the same number of topics, hence showing that the proposed framework can serve as an alternative option.

In addition, our study succeeds in linking the topics that characterize security threats of the past with more recent vulnerability entries, as the extracted topics that were assigned to these entries were based on the content of the vulnerability records from 2015 to 2021. This observation reveals that many, if not all, weaknesses that were detected in the past are still threatening systems and applications and will possibly be repeated in future disclosures as well; otherwise, the topic distributions of the most recent vulnerabilities (2022 entries) would not provide significant information and negatively affect our results. Finally, by exclusively using recent vulnerability records to train classification models that are evaluated with acceptable accuracy metrics, we also discover that the entries of 2022 are related to each other in terms of exploitation and textual information.

Regarding the practical use of these findings, we believe that several key points can be considered significant for researchers and engineers. For example, machine learning engineers who aim to optimize the process of vulnerability prioritization for a company can leverage the above results and consider the proposed framework as a good alternative for assessing and classifying future security threats. Also, since our experiments were completed using only the short descriptions of vulnerabilities, our research encourages engineers in evaluating the exploitability of vulnerabilities from the moment they are disclosed without requiring additional features and time. Finally, both researchers and engineers can leverage our findings to reduce the requirements of their implementations in vulnerability assessment, as our experiments reveal that the descriptions of a relatively small and recent dataset can effectively predict and assess the exploitability of future vulnerabilities.

## 5. Discussion

To recap, we notice that the proposed framework (PF) extracts clusters from a large corpus that can effectively capture keyword relations and unveil coherent topics, making this approach a productive option for future research attempts. Also, in contrast to most approaches of this nature, this approach offers visualization capabilities and exports outcomes significantly faster. During the experiments of this study, we noticed that the process of training all the examined models using the PF was finalized significantly faster than the two topic modeling algorithms. However, we cannot determine which algorithm has the lowest time complexity/costs as the running time of each algorithm strongly depends on the initial model parameters and stopping criteria, which raise bias in the process by affecting the number of iterations.

Regarding the space and time costs of the proposed method, firstly, the cost of the GloVe algorithm is $O\left(\text{no\_keywords}^2\right)$ [12] as it depends on the keyword co-occurrence statistics (TCM). Moreover, the respective cost of UMAP is empirically approximately $O\left(\text{no}_{\text{keywords}}^{1.14} \times \text{no\_UMAP\_dimensions}\right)$ [15], where no_UMAP_dimensions denotes the number of dimensions of the outcoming vectors; in our case, it was set to two. Finally, the cost of the FKM algorithm is $O(\text{no\_keywords} \times \text{no}_{\text{clusters}}^2 \times \text{no\_UMAP\_dimensions} \times \text{no\_iterations})$ [71].

Fitting a topic modeling algorithm is an iterative process where the linkage strength between every single keyword—included in the documents—and every predefined topic is re-evaluated continuously. Thus, the complexity of the two topic modeling algorithms can be considered as $O(\text{no\_documents} \times \text{no\_keywords} \times \text{no\_topics} \times \text{no\_iterations})$ [23]. By inspecting the costs of these two different approaches, we can conclude that the PF requires

fewer memory resources and probably fewer computations per iteration than the two topic modeling algorithms.

Overall, and most importantly, we should mention that most of the topics, extracted from the PF, are matched with basic CWE identifiers and descriptions as well as with some software products and vendors. Also, by considering the findings discussed in the previous section, we finally reach a point where the topic properties discovered from the PF can both effectively characterize new entries and predict their exploitability, helping us to satisfy both $RQ_1$ and $RQ_2$. Moreover, by proposing a new approach that produces high performance for both topic and classification purposes, we succeed in boosting the accuracy and significance of the findings that address the posed tasks of this study.

The outcomes of this study contribute to providing both practical and informative knowledge as we both reveal topic details of security vulnerabilities and present a new structured framework. Briefly, this framework combines several methodologies in a serialized way that are utilized in both general tasks and text mining approaches. Undoubtedly, both UMAP and FKM are algorithms that provide sufficient results in various tasks while GloVe constitutes a validated approach in projecting keywords in vector spaces. These three algorithms offer analogous qualities to the proposed framework in the stage of topic extraction. In particular, GloVe produces a multidimensional vector space that reflects on the relations between the collected keywords while UMAP filters the latter vector space with an ultimate goal to gather and spread the keywords appropriately. At last, FKM identifies the suitable clusters included in the outputted 2d vector space, helping us to discover and interpret intelligible topics as well as assign topic distributions over the documents of the corpus, i.e., descriptions of security vulnerabilities.

By analyzing the performance of the three examined approaches, in terms of topic coherence and classification power, we cannot determine with high certainty whether one of these approaches overshadows the rest. However, we can clarify that CTM provided lower predictive power than PF and LDA and that PF provided the highest performance in relatively many cases. With more detail, we evaluated nine models overall, three for each algorithm, where the three models that were trained based on the PF were evaluated as the most coherent ones. Compared with the highest evaluations extracted using the two topic modeling algorithms, the PF improves the best solution by 20% for 10 topics, 45% for 21 topics, and 55% for 24 topics. Also, the PF provides the best evaluations eight times out of fifteen compared with the other classification models that were trained under the same number of topics, while three out of the five metrics are maximized for the model that was trained for 24 topics using the PF.

Regarding the exploitability of security vulnerabilities, we reveal that the vulnerabilities that are related to specific types of weaknesses or products are more likely to be exploited. To be more precise, the information extracted from the recent vulnerability records indicates that the most exploitable types of weaknesses are SQL injection along with buffer and stack overflow. At the same time, we discovered that the security vulnerabilities associated with WordPress plugins and the TensorFlow learning platform tend to be more exploitable than the ones associated with other products. Hence, we propose that the aforementioned types of vulnerabilities and products should be considered as the main priorities of users and experts in terms of avoiding potential security breaches as well as maintaining the security of applications and systems. The final GLM shows that only three out of the twenty-four topics are not assigned with statistically significant coefficient estimations, which means that the extracted topics offer sufficient information on the exploitability of security vulnerabilities. The coefficients of the GLM show that SQL injection constitutes the most severe one as the odds of exploitation have an estimated relative increment of more than 400 for a unit increment. At the same time, the buffer/stack overflows, the WordPress plugins, and the TensorFlow learning platform are linked with close to 172, 32, and 57 estimated relative increments to the odds, respectively.

Overall, these estimations reveal that the majority of the extracted topics are associated with a small proportion of exploitable vulnerabilities as only five of them produce positive

and significant coefficients. In addition, our experiments show that only twelve topics are associated with significantly low estimations. Therefore, we suggest that the vulnerabilities associated with the remaining topics should be accounted for as potential severe threats since the available technologies and capabilities evolve and change over time.

## 6. Threats to Validity

Although we believe that the results of this study are promising, we suggest that additional experiments are necessary as they could boost the performance and validate the effectiveness of our framework. The issues we encountered are linked to both topic extraction and classification tasks; they generally concern model tuning, exploration of additional algorithms, performance evaluation, results interpretation, and data collection. As a result, the main drawbacks and limitations of this study are related to both the internal and external validity of this study.

Regarding the internal validity, we first introduced some methodologies for data collection, cleaning, and preprocessing that contain some manually treated steps, especially the ones that are related to the main annotation. Although the related literature proposes several approaches and data sources to characterize each vulnerability as exploitable or not, many of the respective studies propose the NVD as one of the primary sources for retrieving information about security vulnerabilities. In addition, the evaluation of the clustering and topic models was completed using an individual measurement, meaning that other characteristics and aspects of these models were not examined. To address this issue, we made sure that the extracted topics were interpretable and validated from the descriptions and properties of official vulnerability records; further, we proceeded to visualization techniques that provided additional information. We also made sure that the employed evaluation metric, i.e., NPMI, is widely applied and supported by the related literature. Finally, the tuning procedures of the classification models are a potential threat to many tasks, which are similar to the ones included in this study, as there is not yet an overall optimal approach to deal with this challenge. For this reason, we employed a validation strategy that explores the performance of classification models under different combinations of parameters to select the most productive ones from a large pool of models.

The external validity of this study concerns both the generalization of our findings and the effectiveness of the proposed framework on different data structures. First of all, to generalize our findings on the most exploitable threats and products, it is necessary to retrieve and analyze collective knowledge from multiple sources, which are possibly not accessible for security reasons, as a single one may not cover every single relevant event, clue, or proof. Nonetheless, by evaluating the available information from NVD, we succeed in including valuable information from multiple data sources since the external references of NVD are linked to many valuable security-oriented websites, including security advisories and the ExploitDB. Moreover, apart from security vulnerabilities, one different aspect of cybersecurity includes the malware products that are utilized to perform a series of actions in an affected system. However, security vulnerabilities are usually associated with this aspect as the malware products aim to exploit specific weaknesses in order to gain access to a system. Therefore, we believe that a valuable perspective of computer security is addressed in this study as we reveal the current primary threats that are related to security vulnerabilities. Finally, we believe that the effectiveness of the proposed framework should be investigated in different datasets to accept this option as a productive alternative to topic models. To mitigate threats of this nature, in this study, we decided to employ machine learning techniques that were previously proposed as practical options in various text mining tasks.

## 7. Conclusions

In this study, we made use of multiple machine-learning algorithms that are related to text mining, topic modeling, word embeddings, classification, and cluster analysis. Our goal was to establish models that could be useful in vulnerability prioritization and provide

insights concerning recent exploitable vulnerabilities. The ground truth and main information of security vulnerabilities were retrieved absolutely from NVD data feeds. Therefore, the combined contributions of our study are both informative and methodological, as we both provide valuable information concerning security vulnerabilities and deploy a novel framework to achieve our goals. This framework relies on keyword vectors followed by fuzzy clustering and is further evaluated based on its comparative performance to standard topic modeling algorithms. The corresponding results suggest that this approach stands as an effective alternative for satisfying topic extraction and classification tasks.

The main idea of this study was to focus on the exploitability properties of recent vulnerabilities as the trends of security weaknesses and threats change over time. Thus, among the detected topics, we distinguished the most exploitable and non-exploitable themes on the late disclosed vulnerabilities by using some appropriate methodologies. By doing so, we advise both individuals and organizations to focus on mitigating specific types of vulnerabilities and maintaining the security of some highly exploitable products. To summarize, we succeeded in addressing both RQs and the general goal/challenge of this study. This was achieved via an analytical framework that can characterize future vulnerabilities with reasonable topics and predict their exploitability based on these topic properties.

## 8. Future Work

After reviewing all the discussed threats and potential improvements, we first recommend further experimentation on the proposed framework in corpuses of different natures. We believe some alternatives that apply to this framework could be more effective in future experiments and with different data. For example, instead of GloVe, word2vec, pre-trained word vectors, or methods for document vectors could replace the keyword and document representations of the proposed framework. Moreover, alternative approaches in the clustering phase, like GMM instead of FKM, and in the classification phase, like GBM instead of RF and C5.0, are also recommended as worthwhile modifications.

Overall, we strongly support that model selection and tuning procedures should be revised and re-examined in experiments that include tasks of this nature. Finally, to provide a more general overview of potential threats, we propose that future studies should focus on acquiring information from multiple data sources that cover different aspects of information security. These aspects could be related to both the perspective of an attacked system (countermeasures, vulnerabilities) and the capabilities of potential attackers (malware techniques).

**Author Contributions:** Conceptualization, K.C. and L.A.; methodology K.C. and N.M.; software, K.C.; validation, K.C., N.M. and L.A.; formal analysis, K.C. and N.M.; investigation, K.C. and N.M.; resources, K.C. and N.M.; data curation, K.C. and L.A.; writing—original draft preparation, K.C.; writing—review and editing, K.C., N.M. and L.A.; visualization, K.C. and N.M.; supervision, N.M. and L.A.; project administration, L.A.; funding acquisition, L.A. All authors have read and agreed to the published version of the manuscript.

**Funding:** This research received no external funding.

**Data Availability Statement:** Data and scripts (100% in R) are available in a publicly accessible repository: https://github.com/koncharman/Vulnerability_exploit_prediction_using_descriptions (accessed on 11 July 2023).

**Acknowledgments:** The research in this paper is part of the PhD dissertation of the first author and is not supported by any external funding.

**Conflicts of Interest:** The authors declare no conflict of interest.

## Appendix A

**Table A1.** Major abbreviations.

| Abbreviation | Description |
| --- | --- |
| POC | Proof Of Concept. We refer to this abbreviation when we discuss Exploit Proof Of Concepts. |
| NVD | National Vulnerability Database. The main data of this study were downloaded from their data feeds. |
| RF | Random Forest machine learning algorithm. It was employed in our experiments for exploitation prediction. |
| bow | Bag of words term weighting scheme representing the raw term frequency of each keyword in a document. This scheme was employed to finalize LDA and the DCM. |
| DTM | Document-Term Matrix. An appropriate document representation for topic modeling. |
| GloVe | Global Vectors machine learning algorithm. Helped us establish word representations in a multidimensional embedding space. |
| UMAP | Uniform Manifold Approximation and Projection for dimension reduction. |
| nn | The number of nearest neighbors (prior parameter for UMAP) that are considered in projecting the topology of a data point in a low-dimensional space. |
| FKM | Fuzzy K-means algorithm. It was employed to cluster keyword vectors extracted from UMAP and to later interpret topics. |
| LDA | Latent Dirichlet Allocation. A standard topic modeling algorithm that was used to compare the efficiency of the proposed approach that is related on keyword clustering. |
| CTM | Correlated Topic Models. A standard topic modeling algorithm that was used to compare the efficiency of the proposed approach that is related on keyword clustering. |
| VEM | Variational Expectation Maximization algorithm. This algorithm was followed as a basis to define stoppage criteria for the topic models (LDA and CTM). |
| NPMI | Normalized Pointwise Mutual Information. This measure was used to evaluate the topic coherence of the topic and clustering models in capturing the semantics of the dataset. |
| k | Number of clusters and topics. |
| $U - u_{ig}$ | Posterior cluster (g) memberships of the keywords (i). |
| $H - h_g$ | Topology of the cluster (g) centers |
| $X - x_i$ | Initial vector space inserted in the Fuzzy K-means algorithm |
| KDP | Keyword Document Presence. The number of documents that contain a keyword at least once. |
| KTLS | Keyword–Topic Linkage Strength. Linkage strength calculated by multiplying KDP and U. It is used to identify the top words of each cluster and topic |
| RS | Row Sums refer to the numerical sums of the elements included in each row of the DTM. |
| DCM | Document Cluster Membership. It denotes the matrix that stores the memberships linking each document with clusters extracted from a FKM model. |
| AUC | Area Under Curve. A standard performance evaluation measure of classification machine learning models. |
| PF | Proposed Framework. It is first introduced in the second section of the presented results and refers to the complete framework that combines GloVe, UMAP, and FKM. |

**Table A1.** *Cont.*

| Abbreviation | Description |
|---|---|
| GLM | Generalized Linear Models. In our study, a model of this nature was employed to address the potential effects of the extracted topics on a target class. |
| $b_j$ | Coefficient of the *j*-th predictor in a GLM. |
| EDF | Expected Document Frequency. This abbreviation refers to the expected (mean) cluster memberships of the latest documents (2022) included in this study. |

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
