# Peer review of "Exploitation of Vulnerabilities: A Topic-Based Machine Learning Framework for Explaining and Predicting Exploitation"

_information, doi:10.3390/info14070403_

Round 1

Reviewer 1 Report

The research study is about the Exploitation of Vulnerabilities: A topic-based machine learning framework for explaining and predicting exploitation. The research work is somehow new, but several limitations are the technical hinder to grasping the main theme of the work, some of them are listed below:

1.      Please justify the importance of software vulnerability prioritization adequately in the introduction section.

2.      How does the proposed framework contribute to the existing research on software vulnerability prioritization?

3.      Please discuss the methodology for topic extraction and classification models in detail.

4.      How does the unsupervised algorithm Global Vectors perform in constructing word representations?

5.      Is the dimensionality reduction technique (Uniform Manifold Approximation and Projection) effectively applied in the research?

6.      How does the Fuzzy K-Means algorithm perform in assigning cluster memberships and topic distributions?

7.      The discussion section is overly descriptive and lacks critical analysis. Please discuss the experimental results in a clear and comprehensive manner.

8.      How do the proposed topics extract from vulnerability descriptions compare to existing topic modeling techniques in terms of coherence and accuracy?

9.      Are there any limitations or potential biases in the data sources that should be addressed?

10.  What are the implications and potential applications of the research findings in the field of software vulnerability prioritization?

11.  What are the limitations of the proposed method? Are there any potential drawbacks or shortcomings that need to be addressed?

12.  In the discussion section, the authors must talk about the computational overhead in terms of the cost and complexity of the proposed method.

13.  Do the authors use any cross-validation techniques for the performance evaluation?

14.  Please separate the future work from the Conclusion section.

Reviewer 2 Report

To improve the quality of the paper, the reviewer gives the following comments.

(1) Section 3 should explain why choose the RQ1 and the RQ2. In addition, the definition of “recent exploits” is  unclear.

(2) All symbols in the equations need to be explained. For example, “uij” in equation (2) is not discussed.

(3) Before showing the results in Section 4, it should simply describe the experimental platform.

(4)  It needs to explain how to get GLM coefficient estimate In Table 5.

(5) In Table 6, the meaning and the computation of AUC is unclear.

The writing quality of the paper should be improved. The editing errors include “Exploitation of Vulnerabilities: A topic-based machine learning framework for explaining and predicting exploitation” in lines 2-3, “Text mining” in line 27, “studies indicate that FKM have been” in line 81, “propose an exploitation scoring system that instead of assigning binary classes that indicate the likelihood of vulnerability exploitation, assigns probabilities of exploitation and provide detailed information for vulnerability prioritization” in lines 113-116, and “we make use the extracted processed” in line 221, etc.

Reviewer 3 Report

In this paper, authors proposed a framework that maps newly disclosed vulnerabilities with topic distributions and further predicts whether this new entry will be associated with a potential exploit Proof of Concept. The idea of this paper is interesting. I have some comments:

1.      The abstract of the paper is vague.

2.      Authors mentioned that “the prioritization of software vulnerabilities is associated with its likelihood of exploitation”, I agree with this, but prioritization of the vulnerability can also be dependent on the severity of the vulnerability, complexity of the attack, value of the assets, and its impact.    

3.      Authors mentioning the “software vulnerabilities” and using NVD data base. NVD includes all the security vulnerabilities related to the software, hardware, and network. Did the authors only used the software vulnerabilities data? It is not clear.

4.      Authors mentioned that they “predicts whether this new entry will be associated with a potential exploit Proof of Concept (???)”, which databases are used to find potential exploit? Is it only NVD or other databases are used?

5.      In section 4.2, results are not clear that how they did exploit prediction, discuss in detail that how the results can be used by the researchers and engineers?

6.      In table 1, is it good to write “Textual description”.

7.      In equation 8, authors have used “2”, is it correct?

No

Round 2

Reviewer 1 Report

After carefully reviewing the revised version of the paper, I recommend its publication. The authors have adeptly addressed my concerns, significantly improving the clarity of their work. The study's findings hold substantial implications, making it a valuable addition to the scientific literature.

Reviewer 2 Report

Eq. 8 is the formula to compute F1 not AUC. In fact, the reviewer still does not know how to compute AUC.

“Exploitation of Vulnerabilities: A topic-based machine learning framework for explaining and predicting exploitation” in lines 2-3 should be “Exploitation of Vulnerabilities: A Topic-Based Machine Learning Framework for Explaining and Predicting Exploitation” and “Text mining” in line 25 should be “Text Mining”. The writing quality needs to be improved.

Reviewer 3 Report

Authors have made significant changes in the article based on my comments. The paper looks good now. 
